# From Molecular Understanding and Pathophysiology to Disease Management; A Practical Approach and Guidance to the Management of the Ocular Manifestations of Cystinosis

**DOI:** 10.3390/ijms26178237

**Published:** 2025-08-25

**Authors:** Hong Liang, Christophe Baudouin, Bénédicte Dupas, Thibault Delcroix, Vincenzo Giordano

**Affiliations:** 1Hôpital National de La Vision des 15-20, INSERM-DGOS CIC 1423, IHU FOReSIGHT, 75012 Paris, France; bdupas@15-20.fr (C.B.); cbaudouin@15-20.fr (B.D.); 2Institut National de La Santé et de La Recherche Médicale INSERM UMRS 968, CNRS, UMR 7210, Institut de La Vision, IHU ForeSight, Sorbonne Université UM80, 75012 Paris, France; 3Recordati Rare Diseases, 52, Avenue du Général de Gaulle-92800 Puteaux, 75012 Paris, France; delcroix.t@recordati.com (T.D.); giordano.v@recordati.com (V.G.)

**Keywords:** cystinosis, ocular, inflammation, vicious cycle, guidelines

## Abstract

Cystinosis is a rare lysosomal storage disease characterised by cystine crystal formation within cells. In the eyes, crystals accumulate in the cornea causing photophobia, loss of visual acuity, and corneal complications. Strict adherence to topical cysteamine treatment is the only therapy that reduces corneal crystal accumulation. Cystinosis, a crystallopathy, is also a disease of inflammation. As the disease progresses the inflammatory processes have a greater impact on the ocular manifestations. The age at which inflammation becomes increasingly significant is dependent on the adequacy of early patient management and adherence with therapy. As patients are living longer with cystinosis, optimising ocular management is increasingly important. No clinical guidelines addressing the long-term ocular management of cystinosis exist. Similarly, there is little recognition in the literature of how to address the inflammatory component of the disease. This paper presents management guidelines, linked to the 3C Classification of severity, used at our centre that provides a framework for optimising care. Adoption of these can help preserve the sight of cystinosis patients. The paper also hypothesises the molecular pathway leading to corneal inflammation.

## 1. Introduction

Cystinosis is an autosomal recessive disease characterised by cystine crystal formation within the lysosomes of cells. It can be categorised as both a crystallopathy, and a lysosomal storage disease and has a significant inflammatory component to its expression and progression. Crystal formation occurs throughout the body, crystal accumulation with resulting organ damage is seen early in the kidneys [1]. The proximal tubule cells of the nephron have limited regenerative capabilities, which contributes to the early onset of Fanconi syndrome and renal impairment [1,2]. In contrast, the cornea is highly regenerative [3,4], which likely delays the more severe clinical manifestations of cystinosis-related corneal damage.

Over time in cystinosis patients, crystals accumulate in organs throughout the body including the liver, thyroid, pancreas, muscle, brain, and eyes [1]. Crystals appear early in the eyes, and this characteristic is still used as a diagnostic confirmation of the disease. Cystine crystals are found in all ocular structures; the most frequently described ocular manifestation of cystinosis is crystal deposition in the conjunctiva and cornea [5,6]. Corneal crystals may appear as early as 6 months and typically are observed between 1 and 2 years old across all corneal layers. By age 10, without treatment, the entire stromal and epithelial periphery may contain crystals. Pigmentary retinopathy can be observed as early as 5 weeks of age, though symptoms often emerge later as crystal accumulation increases [7].

The earliest and most common ocular symptom is photophobia, it may present in patients before age 5 and rapidly impair the patient’s quality of life [5,6]. By age 10, photophobia is observed in 50% of patients and blepharospasm may present [8].

From puberty, superficial punctate keratopathy and macular abnormalities can be observed alongside increased eye pain and foreign body sensation. By the second or third decade significantly reduced the best corrected visual acuity (BCVA), reduced colour, peripheral, and night vision, and retinal blindness can occur [9]. Band keratopathy, keratitis, neovascularisation, and corneal ulceration become more prevalent as the disease progresses [7].

Improvements in renal transplantation and management mean more patients are living into adulthood, increasing the prevalence of progressive ocular manifestations [10]. Cystinosis most commonly manifests in the vascular posterior segment of the eye as retinal pigment epithelial mottling, which can ultimately lead to reduced night and colour vision. It can also cause moderate to severe constriction of the visual field and moderate to severe reduction in rod and cone mediated ERD responses in older patients. Retinopathy frequencies have been found to inversely correlate with time receiving systemic cysteamine therapy [11]. While the routine examination of the eye of a cystinosis patient includes assessment of the posterior section, this paper is focused on the management of anterior segment complications.

Systemic cysteamine therapy does not treat the avascular corneal structure. Topical cysteamine therapy is the primary treatment for corneal cystine accumulation, it dissolves corneal crystals and alleviates symptoms at all ages [5,7,12]. Early treatment with, and strict adherence to both topical and systemic cysteamine therapy significantly improves the long-term prognosis of ocular cystinosis [10]. Topical ocular cysteamine therapy should be initiated under the supervision of a cystinosis specialist [10,13,14,15].

Individuals with cystinosis face both cystinosis-related eye damage and common ocular issues seen in the general population. Some of these may be exacerbated by conditions such as acute inflammation or dry eye disease (DED). Understanding which eye problems can be treated alongside corneal cystinosis therapy is crucial. While a standardised ophthalmological assessment for cystinosis patients has been proposed [16], no clinical guidelines currently address the long-term ocular management of cystinosis.

It can be seen from this list of ocular symptomologies that early signs and symptoms are crystal-based and complications occurring at a later stage have a larger inflammatory component. Photophobia which occurs relatively early in ocular disease is thought to be caused by light scattering induced by crystals. However, Liang et al. in 2015 also showed that the number of dendritic inflammatory cells correlated to both subjective and objective scores of photophobia [17]. Meanwhile inflammatory cell density correlated to crystal density and nerve damage. Some patients with cystinosis had 16 times the expected inflammatory cell density in the central part of the cornea and these eyes had the highest score for photophobia and nerve alterations [17]. As the disease progresses, processes such as neovascularisation and band keratopathy occur and the inflammatory impact of cystinosis increases.

### 1.1. Cystinosis—A Crystallopathy

Crystallopathies are harmful states or diseases associated with the formation and aggregation of crystals in tissues or cavities [18,19]. They are caused by intrinsic or environmental microparticles or crystals and these promote tissue inflammation and scarring [20]. Supersaturation at a local level has been considered as a common trigger for crystallisation, and upon formation of a nucleus, the crystals self-perpetuate and cause further aggregation and crystallisation [18].

In proliferating cells, lysosomal concentrations of amino acids are typically lower with one exception: cystine, the dimeric form of cysteine. Cystine is ~30-fold more concentrated in lysosomes compared to the cytosol [21]. In cystinosis, the role of cystinosin is deficient and this high cystine concentration may explain in part the early development of visible crystals in the body in infantile or nephrotic cystinosis where the deficiency is most severe.

Crystallopathies have been linked to the production of reactive oxygen species, immune cell recruitment and activation, and increased expression of various inflammatory cascade molecules [22]. The process of crystallisation elicits direct cytotoxic effects, starting an auto-amplifying inflammatory loop commonly ending in cell death [18].

Mulay et al. (2016) characterise three methods by which crystallopathies cause disease, namely by acute necroinflammation, mechanical obstruction and chronic tissue remodelling [18]. Necroinflammation is described as an auto-amplifying process where crystals are cytotoxic and cause cell death with a local and systemic inflammatory response [18].

Crystals can also directly activate inflammation, through many pathways, calcium and potassium signalling, calpains, cathepsin beta, proteases, and inflammasomes including the NOD-, LRR- and pyrin domain-containing protein 3 (NLRP3) inflammasome [23,24,25,26,27,28].

Cell death induced by a crystallopathy can be by apoptosis, or necroptosis—an inflammatory cell death. When cells become stressed, they secrete alarmins, proteases, and damage-associated molecular patterns which further activate Toll-like receptors and inflammasomes. These activate the inflammatory response including the release of interleukins, cytokines, kinins, lipid inflammatory mediators, complement system activation, vasodilation, an increase in endothelial permeability, and leukocyte influx. Macrophages attempt to remove crystals from tissues by phagocytosis. If digestion of the crystalline material fails in the lysosomes, however, macrophages undergo autophagy [18].

The most straightforward treatment of crystallopathies would be to disrupt or dissolve the crystals. In cystinosis this is achieved through cysteamine therapy. The primary mechanism of action of cysteamine hydrochloride involves its ability to enter lysosomes and react with cystine to form cysteine and cysteine-cysteamine mixed disulfides. These products can then exit the lysosome, reducing the overall cystine load within the cell.

### 1.2. Cystinosis—An Inflammatory Disease

Inflammation has recently been revealed as a major contributing mechanism to the pathogenesis and progression of both the renal and systemic involvement in cystinosis [29,30]. Cystine crystal accumulation, macrophage activation, enhanced oxidative stress, and inflammasome pathway activation are all driving factors behind the vicious cycle of cellular inflammation in cystinosis, which, no matter what the affected organ is, commonly ends in tissue fibrosis and loss of function.

In cystinotic patients and in *Ctns*−/− mice, cystine crystals have been observed in the macrophages of most organs, including bone marrow, kidneys, liver, skin, and gastrointestinal mucosa [31,32,33,34]. Cystine crystal accumulation is in part due to the dysfunction or absence of cystinosin, and in part due to the phagocytosis of the dying cells releasing their intracellular contents including cystine crystals. Cystinotic macrophages cannot get rid of the phagocytized cystine crystals because they have the same genetic and biochemical defect as other cytinotic cells, thus they tend to enter a non-ending cycle of cystine crystal deposition, wherein they send signals for the recruitment of other inflammatory cells and finally die [29].

In *Ctns*−/− mouse kidneys, macrophages mainly display an M1-like pro-inflammatory profile [35]. This helps to explains why cystinotic patients have significantly elevated plasma levels of interleukin-1β (IL-1β) and an increase in IL1β mRNA expression levels in the peripheral blood mononuclear cells [30]. Cystinosin, independent of the accumulation of cystine crystals, also leads to macrophage-mediated inflammation in cystinosis by regulating the localization and degradation of the β-galactosidase-binding protein, galectin-3. Galectin-3 is an inflammatory mediator that plays a role in acute and chronic inflammation by attracting monocytes/macrophages in diseased tissues [35].

Although the exact nature of the immunostimulatory signals activating macrophages in cystinosis is still unknown, it was hypothesised by Elmonem et al. that cells damaged by the crystals release damage-associated molecular patterns (DAMPs) that, in turn, initiate inflammatory responses by triggering various cells, including tissue-resident macrophages, and result in the release of pro-inflammatory cytokines and chemokines—such as IL-1β, IL-6, tumour necrosis factor-α (TNF-α), and monocyte chemoattractant protein-1 (MCP-1). The released cytokines and chemokines promote the recruitment of inflammatory cells that contribute to the exacerbation of the tissue injury [29].

Inflammasomes are also activated in cystinosis. These multiprotein complexes play a major role in innate immune responses by controlling the caspase-1-dependent proteolytic maturation and release of the pro-inflammatory cytokines IL-1β and IL-18 [36]. Cystine crystals activate the inflammasome system in control of the monocytes and induce caspase-1-dependent IL-1β secretion through a mechanism involving cathepsin B leakage, reactive oxygen species (ROS) production, and potassium efflux. In cystinotic patients, the circulating levels of the inflammasome-induced cytokines IL-1β and IL-18 were found to be significantly elevated when compared with those observed in healthy subjects [30].

The relationship between inflammation, autophagy, and apoptosis is altered in cystinosis. Autophagy is a cellular process involved in the restoration of energy homeostasis through the catabolism and recycling of dysfunctional proteins and cellular organelles. It improves cell survival upon exposure to various stress conditions. Whether cells survive during inflammation is dependent on the balance between the pro-survival mechanisms of autophagy, when functioning properly, and the pro-death mechanisms of apoptosis or necroptosis [37].

In cystinosis, structurally and functionally abnormal mitochondria are an important aspect of the disease. As a major source of ROS, mitochondria are particularly susceptible to oxidative stress resulting from the inflammatory processes triggered by cystine accumulation. This usually leads to the abnormal induction of mitochondrial autophagy, and this is further associated with reduced mitochondrial ATP generation, cell starvation, increased generation of ROS, autophagic flux blockade, and enhanced apoptosis [38].

A recent study stated that the primary biochemical/physiological defect in cystinosis is failure to supply cysteine to mammalian target of rapamycin (mTOR) via cystinosin [39]. Cystinosin interacts with mTOR [40,41], and recently this was described as playing a major role in cell fate in cystinosis [42]. In immune cells mTOR regulates metabolism to fuel cell fate decision, proliferation and effector functions. In non-immune cells, it controls inflammation-associated proliferation and migration/invasion, shapes the expression of cytokines and chemokines, and promotes extracellular matrix remodelling and fibrosis. mTOR plays a critical role in chronic inflammation, where continuous feedback between stromal cells and infiltrating immune cells results in tissue remodelling and organ damage. It has been noted that a wide range of the aberrant cell physiology in cystinosis can be explained by the disrupted mTOR pathway [39].

Importantly, in acute inflammation, mTOR acts as a pro-inflammatory mediator, increasing nuclear factor kappa B (NF-κB) activity and ROS production [43,44]. However, in later stages of cystinosis, due to the upstream defect in cystinosin-mediated cysteine transport, mTOR is no longer adequately activated. Therefore, mTOR loses its ability to mediate NF-κB and ROS signalling.

Most of this work has been performed with reference to kidney function or using models of the disease based on kidney function. However, the underpinning inflammatory processes in the eye are likely to be driven in a similar way.

### 1.3. 3C Classification

In 2025 a new categorisation for ocular cystinosis patients known as the 3C classification (crystal-complication-compliance) was published [45]. This classification recognised both the impact of crystals and also that of complications driven by both crystal accumulation and inflammatory processes such as keratitis, neovascularisation and band keratopathy. The third part of the classification sought to understand the level of compliance with cysteamine therapy. This classification is based on expert ophthalmologist observation and showed good replicability of the categories by linear determinant analysis.

The four categories used in this classification can be broadly summarised as follows [45]:

Category 1: Patients with evidence of crystals but low to moderate crystal accumulation scores. No loss of BCVA, mild or no photophobia. No corneal complications. Patients are generally young (<20 years old) with good compliance to ocular therapy likely due to parental assistance.

Category 2: Patients with evidence of moderate crystal accumulation, optical coherence tomography (OCT) %, representing crystal deposition depth as a percentage of corneal thickness, is likely to be >30%. Photophobia is more prevalent although still mild to moderate, BCVA remains largely unaffected. No ulceration or band keratopathy but mild keratitis and neovascularisation can present. Patients are usually 10–30 years old with lower treatment compliance than patients in category 1. This is likely due to the transition from parental to self-instillation and a lack of awareness or concern regarding the consequences of crystal accumulation.

Category 3: Patients show extensive crystal accumulation with an OCT% score > 50%. Photophobia is moderate with all patients ≥ 1 and likely 2–3 on the Liang 5-point scale (where Grade 0 is no photophobia under the slit-lamp beam and grade 5 is where a patient is unable to open their eyes even in a darkened room) [17]. BCVA is affected in most patients although not all. 90% of patients have band keratopathy. Patients are likely to be 20–45 years old with compliance improving compared to category 2 as they are now more aware of the ocular damage caused by cystinosis.

Category 4: Crystal deposition is widespread with an OCT% score > 80%. Patients typically experience moderate to severe photophobia, with 83% scoring at least 3 on the Liang 5-point scale. BCVA is moderately to severely affected. All patients have corneal complications, 75% experience all four complications routinely assessed (keratitis, band keratopathy, ulceration, neovascularisation). Compliance is the lowest in this group due to the severity of the disease making instillation increasingly difficult and uncomfortable. Patients at the severe end of category 4 are considered as a subcategory—category 4 late stage. Here medical therapies have little or no effect. The structural integrity of the eye is severely affected by the progressive processes of the disease and patients need to be prepared for loss of sight.

### 1.4. Therapy Adherence and Medical Management

A primary goal of therapy is to keep patients in their 3C classification category for as long as possible. An objective of management is to try and prevent patients from progressing to a higher category, particularly from Category 3 to Category 4, which represents the bridge at which the increasingly progressive inflammatory processes become hard to control and manage. The observations supporting the 3C classification show that there are patients in Category 1 who are in their fourth decade of life, but there are also patients in Category 2 and Category 3 who are in their first decade of life [45]. Adherence is a contributing factor to this, and adherence management is an important part of the clinical management process.

### 1.5. Repair and Healing of the Cornea

Repair and healing of the cornea is complicated and varies by layer. Slight injuries of the central cornea repair on their own but, in general, the epithelium heals through the proliferation of stem cells and their differentiation into specialised epithelial cells and their migration to the damaged area. In contrast, stromal cells undergo transformation, whereas the endothelium regenerates especially via cell migration [46]. The regeneration process is complicated and injury-induced intercellular crosstalk is thought to involved. The most relevant growth factors triggering cell migration are insulin-like growth factor (IGF) 1, epidermal growth factor (EGF), hepatocyte growth factor (HGF), keratinocyte growth factor (KGF), transforming growth factor (TGF), and cytokines, such as IL-1 and TNF-α [47,48,49]. The extracellular matrix (ECM) interacts with this process providing a scaffold for it. Corneal epithelial cells secrete components that form the ECM above Bowman’s layer. If any one of these is deficient, the critical balance breaks down impacting homeostasis and repair.

### 1.6. The Use of Cysteamine and Anti-Inflammatory Agents

The 3C classification was developed at the Department of Ophthalmology III, Quinze-Vingts National Ophthalmology Hospital, Paris where cysteamine eyedrops and ciclosporin are important parts of the treatment armamentarium for anterior eye problems. It is interesting to note that the use of ciclosporin increases markedly by 3C classification category. In category 1 and 2, 4.6% and 6.2%, respectively, received ciclosporin. This rose to 50% in Category 3 and 75% in Category 4. Despite this being common practice in expert centres a review of the literature on the management of ocular cystinosis barely discuss the non-specific treatments and where treatment is mentioned it is primarily focused on the use of cysteamine eyedrops.

Two publications were found that mention the use of anti-inflammatory agents alongside cysteamine. Firstly the “Expert guidance on the multidisciplinary management of cystinosis in adolescent and adult patients” by Levtchenko et al., 2022 states that “complaints related to cystinosis associated ocular surface disease are frequent and can be improved by using artificial tears, anti-inflammatory agents or other local treatment related to the corneal complications due to ocular cystinosis” [15]. The second is the paper on the 3C classification published in 2025 which measures the progression and development of ocular cystinosis in a cohort of 64 patients including the use of ciclosporin and artificial tears [45].

Cysteamine and ciclosporin are complimentary medications. The primary mode of action of cysteamine is cystine crystal depletion and that of ciclosporin is anti-inflammatory. Cysteamine enters lysosomes and reduces the overall cystine load within the cell. These cystine-lowering effects decrease cellular inflammation. It has an antioxidant effect that can suppress inflammatory cascades following oxidative stress. As reported by Okamura et al., it can reduce ROS generation, attenuate macrophage recruitment and activity, block myofibroblast proliferation and activation, and decrease tissue fibrosis [50]. It is also a potent inhibitor of apoptosis in cystinotic cells. This has been attributed to its cystine-lowering effects, linked to its antioxidant and anti-inflammatory properties.

Topical ciclosporin exerts immunosuppressive action on inflammatory mediators by blocking T-cell activation and the release of inflammatory cytokines [51,52]. In the ocular surface epithelium, ciclosporin inhibits apoptosis, preventing the activation of T-cells and the reduction in inflammatory cell infiltrates. The reduction in T-cell recruitment and activation by ciclosporin decreases interferon gamma (IFN-γ) expression, which has been linked to epithelial cell and goblet cell apoptosis [53]. Ciclosporin is used across a gamut of ocular inflammatory conditions. A systematic review analysed 48 randomised controlled trials on topical ciclosporin use in ocular surface diseases and indicated that topical ciclosporin may be more effective in relieving the symptoms and inflammatory impact on the cornea and conjunctiva in ocular surface disease than other frequently used topical therapeutics [54]. Furthermore, the long-term use (6 months) of topical ciclosporin has been found to be safe and effective even in children [55].

Ocular inflammation of the anterior eye may be caused by cystinosis or by other common inflammatory problems like uveitis, allergens, dry eye disease, or scleritis. This inflammation can compound the underlying inflammation caused by cystinosis. It is recognised that as in renal and systemic cystinosis there is a four-phase immune response that comprises initiation, amplification, recruitment, and damage/reinitiation which can result in a vicious cycle of inflammation that does not resolve.

## 2. Recommended Guidance on Managing Cystine Crystals, Inflammation, and Everyday Eye Problems

This guidance is based on the practice of the Department of Ophthalmology III, Quinze-Vingts National Ophthalmology Hospital, Paris and the 3C classification for ocular cystinosis patients [45]. It covers the management approach adopted and practical nuances that support successful patient management and addresses the type of patient, their degree of cystinosis and the impact of other corneal complications on cystinosis management.

### 2.1. Recommended Ophthalmologist Standard Practise for Cystinosis Patients

Figures 2–4 outline the standardised management approach we recommend for the ocular management of cystinosis patients. Our centre uses Cystadrops^®^—the only authorised treatment in Europe—administering one drop four times daily to treat corneal cystine crystal deposits. It is recommended for adults and children from 6 months of age diagnosed with cystinosis and with observed corneal crystal accumulation [56]. Our approach aims to optimise continuous treatment as corneal cystine crystal accumulation is likely to increase if treatment is discontinued and accelerate the progression of the disease and its complications [56].

### 2.2. Routine Cystinosis Assessment and Evaluation

Patient’s eyes should be routinely comprehensively assessed; this includes both the anterior and the posterior segments. At each patient visit a standard protocol is followed based on the French national diagnostic and treatment recommendations for ocular cystinosis designed by our group (Table 1) [57]. Multimodal imaging is crucial for evaluating the impact of cystine crystal deposition across the eye (Figure 1a), while Figure 1b presents an example eye image from each 3C classification category.

Patients are assessed by a slit lamp and the Gahl’s score recorded (from 0.00 to 3.00 in 0.25 increments where 0.00 represents the absence of crystals in the cornea and 3.00 represents a cornea packed with crystals) [5]. OCT is used to observe the cornea, with OCT% recorded for all patients over 3 years old to quantify crystal deposition depth as a percentage of corneal thickness. OCT should also be performed on the macula and optic nerve to assess crystal ingress and cystinotic damage. In our unit, skilled practitioners routinely use In Vivo Confocal Microscopy (IVCM) on patients from age 7 who are comfortable with the procedure. IVCM is recognised as a superior imaging technique for ophthalmological follow-up in cystinosis [15,58].

BCVA and photophobia are recorded. The presence or absence of corneal complications: band keratopathy, keratitis, neovascularisation, and ulceration are routinely noted. The evaluation includes routine assessment of the eye as for patients without cystinosis. Once assessment is complete the patient is categorised according to the 3C classification.

### 2.3. Initiation of Cystadrops^®^ Therapy

Figure 2 outlines the process for initiating therapy. Once a patient has been ocularly assessed for the first time, therapy should be initiated. Therapy should be continuously prescribed unless the patient presents with any of the following: corneal ulcer, corneal thinning, corneal oedema, acute ocular inflammation, severe neovascularisation, or severe band keratopathy. In such cases Cystadrops^®^ should be discontinued and these conditions treated. Upon sufficient stability, therapy should restart.

### 2.4. Optimising Adherence

Adherence, in reference to a patient actively choosing to follow a mutually agreed upon treatment plan, is critical for the success of therapy. We have developed a process to optimise Cystadrops^®^ use and adherence (Figure 3). Adherence encompasses not only following treatment and appointment schedules but also adopting recommended lifestyle changes. We strongly recommend that prescribing physicians possess a thorough understanding and first-hand experience of the instillation process to best support their patients, as this is a critical step in familiarising them with the therapy.

### 2.5. Supporting the First Instillation

Ocular cystinosis patients receiving their first topical treatment are usually young children, though patients may be of any age. Our clinical team considers it essential to sit with the patient and caregivers (as appropriate) during the first instillation to demonstrate the correct technique and assist with any instillation issues that may occur, helping ensure they are not discouraged from continuing therapy.

Cysteamine eyedrop instillation can cause eye pain, eye irritation, blurred vision, and lacrimation increase [13,14,59]. Most of these reactions are transient and mild to moderate. Clinician presence at the first instillation enables reassurance to be provided that any stinging or blurring is transient. It also provides an opportunity to educate, address concerns, and share practical advice such as eyelid cleaning. Educational leaflets are often used to support this process and assist with future instillations. Instillation confidence is a major contributor to therapeutic success.

### 2.6. Facilitating Familiarisation and Comfort with the Installation Process

The clinical team is available, in person or virtually, to address queries during the first few weeks after Cystadrops^®^ is prescribed, helping ensure patients/caregivers are confident in performing the instillation process independently. In our unit, for children under 3, we sometimes recommend that caregivers administer one instillation every 2–3 days until those involved are comfortable with the procedure. While off-label, this approach helps familiarise the patient and caregiver with the administration process gently, helping long-term therapy compliance. Encouragement of both patients and their caregivers at this stage is essential.

### 2.7. Forming Habits

Cystadrops^®^ is indicated four times daily, so maintaining a consistent schedule is crucial for compliance. The clinical team engages with the patient/caregiver to create a timetable that fits the patient’s daily routine. It is vital the clinical team revisits this timetable with patients/caregivers when there are significant life changes such as starting school, moving to independent living, starting a new job, or changes in family circumstance like a new baby.

Category 1 patients are usually young, with their caregivers handling their treatment, which likely explains their good compliance. In Category 2 compliance falls; this correlates with a child’s transition to adolescence and their increased responsibility for self-medication. It is crucial the clinical team are especially attentive to adolescents/category 2 patients to help develop appropriate lifelong treatment habits. The clinical team should support the patient and be available to address questions and concerns in real-time via email, telephone, or other communication methods to ensure continued therapy compliance.

### 2.8. Education Underpins Long-Term Management, Compliance, and Adherence

We believe it is vital that patients/caregivers understand the pathophysiology of cystinosis, including the progressive nature of cystine crystal accumulation which may not initially cause noticeable symptoms. The clinician routinely sharing images, information, and ophthalmological assessments is key to this education. Scan results are shared and compared with previous scans to help patients understand their disease progression. Sharing these test results can provide positive reinforcement to the patient by showing the success of treatment.

Discussions are conducted with patients/caregivers on how to manage difficulties in taking the medicine—for example, a changing daily routine or practical issues like the use of contact lenses, the management of DED, or the desire to wear eye make-up.

The hospital regularly shares information with patients/caregivers about the disease and latest advances in treatment or technologies through face-to-face meetings, social media, app-based interventions, and by communicating through local patients’ associations. Through this approach, the clinical team believes they can enhance confidence and provide psychological support as the patient learns to live with cystinosis.

Compliance using this methodology has been shown to be good. In a study of 63 patients treated at the hospital from 2013 to 2023 in a real-world setting, all were still undergoing cysteamine eyedrop therapy at their last visit [45]. In a trial setting, the Cystadrops^®^ OCT-1 study by Labbe et al. showed that no patients discontinued therapy over the 5-year follow-up of the clinical study [13].

This approach is supported by a meta-analysis of non-adherence studies, which found that two types of intervention can significantly improve medication adherence by patients with adherence problems. Face-to-face interventions and behavioural strategies, such as linking medication administration to existing habits or using cues, are both effective [60].

The purpose of this clinician-to-patient feedback is to empower patients to become experts in their own disease and recognise the long-term damage caused by crystal accumulation. They must take responsibility for self-management and understand the consequences of neglecting it. Ultimately this feedback builds trust and supports goal-setting, with the primary aim of remaining within a 3C classification category for as long as possible.

### 2.9. Ongoing Patient Follow-Up Protocol

The frequency of routine patient follow-up varies by patient 3C classification category (Figure 4). At all follow-up visits a standard eye assessment as for non-cystinosis patients is routinely performed.

Category 1 patients should be seen at least annually. A slit lamp is routinely used as is OCT in patients over age 3 and IVCM over age 7. Eyelid care education is provided, and treatment is initiated as early as possible. The use of Cystadrops^®^ is recommended for patients over 6 months old as soon as crystal deposition is observed.

Category 2 patients require closer follow-up, ideally every 6 months. Routine assessments with a slit lamp, OCT and IVCM are performed at each appointment. The transition from Category 1 to Category 2 is significant and patients need to be educated on the consequences of increasing 3C classification category and further crystal deposition. A major focus is establishing a treatment plan for each patient.

Category 3 patients begin experiencing corneal complications such as keratitis and band keratopathy. Patients should be seen at least every 3 months. Patients need to understand that corneal complications often result from poor compliance and that continued focus on therapeutic compliance is essential, alongside management of the corneal complications. The importance of staying in Category 3 should be clearly understood by the patient.

Patients in Category 4 require additional encouragement and education. Here we see increased inflammation due to increased corneal complications. We recommend follow-up every 3 months, partly due to the need to manage the corneal complications they will experience. Patients need to be prepared for reduced BCVA, partial sightedness and potential blindness. Surgery and other options may need to be discussed.

### 2.10. Managing Concomitant Corneal Problems

Over time many cystinosis patients develop chronic ocular inflammation. Inflammation should be managed with appropriate anti-inflammatory medications such as steroids and ciclosporin; we use these treatments concomitantly with Cystadrops^®^. In our institution we consider it important to treat cystinosis patients with DED, which is often linked to inflammation, with lubricants. The use of prescription only, preservative free eyedrops is mandatory and we avoid using over-the-counter eyedrops to prevent aggravating the condition; if these are not successful, ciclosporin can be used. Only 12.5% of Category 1 patients required artificial tears; this rose to 31.3% for Category 2 patients and 90% of Category 3 patients, highlighting the increasing symptomatic effects of the disease. In Category 4 patients, 75% were using artificial tears but the use of ciclosporin was significant—again intimating greater impact of the complications of the disease.

The incidence of complications increases as the 3C classification category increases. In Category 4, three-quarters of patients had all four common complications routinely recorded in our clinic [45]. In Categories 1–3 it is generally recommended to continue cysteamine therapy to reduce the impact of corneal complications, but Category 4 patients may require a different approach. The corneal surface may become too pitted and porous for cysteamine therapy to be applied, or there may be a need for planned reduction in daily usage. When the cornea is so fragile, the priority is to use as little treatment as possible.

In the case of severe corneal complications, we may stop using cysteamine in the short term. Antibiotic treatment can be used in case of infection. In cases of ulceration, wound healing agents such as vitamin A ointment or scleral lenses can be utilised.

At advanced stages of cystinosis, the regenerative capacity of the cornea gradually declines. Due to the loss of corneal collagen supports, leaving the cornea effectively empty with no internal structure, the cornea’s ability to synthesise and accumulate new crystals declines. Clinically the absence of visible crystals reflects this loss of biosynthetic activity in very-late-stage cystinosis.

The options at this stage are stem-cell-based therapy or corneal transplant, but there is insufficient evidence to make specific recommendations. In the case of corneal transplantation, topical treatment is still necessary because cystinosin-deficient host cells can reinvade the transplanted cornea.

## 3. Discussion

Systemic cysteamine therapy is essential for controlling cystine crystal accumulation in the posterior eye including the macula, while topical cysteamine therapy is essential for arresting the development of corneal crystals [6,7,8]. However, other factors, including other ocular diseases such as DED and inflammation, also play a role, with links between them all. They all need to be considered as part of the management approach.

We have observed and propose there is a vicious cycle of inflammation and crystal accumulation in cystinosis that requires management. Cystine crystals in the cornea induce inflammation, infiltrate and damage corneal nerves, and disrupt the tear film-inducing DED. Meibography assessments show that cystinosis also causes damage to, or the disappearance of, the meibomian glands. Additionally, cystine crystals can infiltrate the limbus, where accumulation leads to limbal stem cell deficiency.

Corneal neovascularisation develops because of hypoxia. Corneal neovascularisation increases in severity due to increasing inflammation. Without treatment neovascularisation will continue to develop and worsen, presenting across the central cornea. As ocular cystinosis progresses severe complications like band keratopathy and ulcerations develop, and finally total opacification of the cornea occurs. This is a cyclical process; the more crystals there are, the greater the inflammation and the greater the consequential problems (Figure 5a,b) [61,62].

Although cystinosis is well-characterised by lysosomal cystine accumulation and its systemic manifestations, the molecular mechanisms driving corneal inflammation, a key component of this vicious cycle remain poorly understood. We propose that the NLRP3 inflammasome pathway—a central mediator of sterile inflammation—is implicated in this process (Figure 6) [26,43,44,63].

In other corneal diseases, including DED and alkali burn injury, NLRP3 activation has been shown to drive inflammation through IL-1β and IL-18 production. For instance, NLRP3 and IL-1β are upregulated in the tears and conjunctival epithelium of DED patients, confirming its activation in human ocular surface pathology [23]. In murine models, NLRP3-deficient mice exhibit significantly reduced inflammation, immune infiltration, and corneal opacity following alkali burns, highlighting the functional relevance of this pathway [24].

NLRP3 activation is typically a two-step process. The priming signal (Signal 1), mediated by NF-κB, induces the transcription of NLRP3 and pro-IL-1β. The activation signal (Signal 2) is triggered by cellular stressors such as potassium efflux, calcium influx, mitochondrial ROS, and partial lysosomal membrane permeabilization [25,26]. Additionally, endoplasmic reticulum (ER) stress can activate NLRP3 through an unfolded protein response (UPR)-independent mechanism [27]. These processes are tightly regulated, as shown by evidence that partial lysosome membrane permeabilization activates NLRP3, while extensive disruption suppresses it [28].

The relevance to cystinosis stems from the fact that several of these NLRP3-activating stimuli are present in cystinotic cells. For instance, cystine crystals can directly trigger inflammasome activation in monocytes, supporting the idea of crystal-induced sterile inflammation [30]. Moreover, cystinotic fibroblasts and renal epithelial cells demonstrate mitochondrial dysfunction and fragmentation, resulting in increased ROS and mitophagy [38]. Lysosomal overload in these cells contributes to ER stress and calcium signalling abnormalities, creating a permissive environment for NLRP3 activations [64].

Thus, despite the current absence of direct experimental evidence in corneal tissue from cystinosis patients, the convergence of these molecular features strongly supports a testable hypothesis: NLRP3 inflammasome activation is a plausible driver of corneal inflammation in cystinosis. We greatly encourage and recommend appropriate studies are commenced to assess the validity of the hypothesis to strengthen understanding and ultimately improve patient care.

In Category 1 patients, with the support of cysteamine therapy the homeostasis of the eye can be largely maintained by reduction in the cystine crystals although there will be initiation of a pro-inflammatory cytokine release. This is amplified through T-cell differentiation and proliferation which recruits and reactivates further immunological responses. In the early 3C Classification categories of cystinosis, there may be a near resolution of the problem. However, as the disease progresses, resolution of the immune response becomes dysregulated, inflammatory processes take over and the vicious cycle accelerates.

Reduction and maintenance of cystine deposition at a low level is critical to mitigate this vicious cycle. Simultaneous use of anti-inflammatory medication such as ciclosporin and steroids may be beneficial. An adolescent boy in our care with a photophobia score of 3, meaning they could not tolerate the blue slit-lamp beam, was treated with a combination of cysteamine and ciclosporin. The photophobia score decreased to 1 after 6 months, meaning they were photophobic to moderate slit-lamp beam light, there was also a decrease in the feeling of irritation upon cysteamine instillation. This illustrates how our guidance can be used to manage patient’s symptoms but also how managing inflammation alongside crystal accumulation as the disease advances has beneficial impact.

It has also been observed that the efficacy of anti-inflammatory agents often decreases in advanced ocular cystinosis again supporting the concept of this vicious cycle and the need to prevent patients’ disease progressing to Stage 4 where the practicalities of treatment are complicated by the condition of the eye and management becomes “palliative” as the eye continues to deteriorate. We recommend combined cysteamine and ciclosporin therapy in ocular cystinosis patients is investigated further to optimise its use and understand how it could limit progression of cystinosis in the anterior eye. In the case of severe corneal complications, we may stop using cysteamine in the short term. Antibiotic treatment can be used in case of infection. In cases of ulceration, wound healing agents such as vitamin A ointment, or scleral lenses can be utilised.

In very-late-stage cystinosis crystal accumulation decreases due to the loss of corneal collagen supports, leaving the cornea effectively empty with no internal structure. The options at this stage are stem-cell-based therapy or corneal transplant, but there is insufficient evidence to make specific recommendations. In the case of corneal transplantation, topical treatment is still necessary because cystinosin-deficient host cells can reinvade the transplanted cornea.

Sometimes patients present at our clinic with corneal complications and ocular surface diseases. With conditions such as corneal ulcer, thinning or oedema, acute ocular inflammatory diseases, and where there is severe neovascularisation or band keratopathy, the recommendation is to avoid or stop cysteamine therapy and deal with these conditions first. When sufficient corneal stability has been restored it is appropriate to start/restart Cystadrops^®^ therapy. It is recommended that the instillation of agents such as ciclosporin or artificial tears occur half an hour prior to cysteamine therapy to allow for best effects. Currently, our use of ciclosporin is empirical and guided primarily by the clinical observations of the treating physician. Assessment of effectiveness is based on inflammation status and, occasionally, photophobia. No specific guidelines for ciclosporin use are currently in place; therefore, we recommend the development of such guidance through collaboration with other expert cystinosis centres experienced in combining ciclosporin with topical cysteamine.

Whilst our centre has experience with Cystadrops^®^ since 2013, we recognise that our guidance recommends patients use the treatment long-term, potentially for several decades. Hospital prepared cysteamine solutions were available for several years preceding this. It is important that the long-term safety of continual administration is fully understood. The Cystadrops^®^ post-authorisation safety study CYT-DS-001 is currently ongoing in Europe. Data collection began in January 2020 and is estimated to continue until the first quarter of 2028. To date no new adverse events have been reported outside those established in the summary of product characteristics and Cystadrops^®^ is considered safe and effective for long-term use [65].

## 4. Conclusions

The presentation of cystinosis in the anterior eye is clinically challenging. Currently there are no guidelines on treatment and very little in the literature on the management of inflammatory aspects of the disease. Only two papers were found that discussed in a perfunctory way the role of anti-inflammatory medication in cystinosis alongside topical cysteamine treatment. Indeed, the use of specific treatments for corneal cystinosis alongside non-specific treatment for coexisting complications is negligible. However, for patients with cystinosis and the ophthalmologists who treat them, the management of normal ocular complications and those caused or exacerbated by cystinosis are part of everyday life.

Whilst the role of cysteamine has been well established in papers by Liang et al. [8,14] the role of other concomitant and supporting therapies is not defined. The publication of the approach adopted at the Department of Ophthalmology III, Quinze-Vingts National Ophthalmology Hospital, Paris represents a significant step forward in terms of supporting clinicians and patients tackle the disease. It sets out a benchmark for the management of the ocular manifestations of cystinosis that

Recognises that management of the condition is throughout the patient’s life.Has a standardised approach to assessing and monitoring the disease that helps to inform clinical decision making.Understands that the management of the disease requires an approach to reduce both crystal accumulation and inflammatory processes, particularly as the disease advances.Recognises that cystinosis can exacerbate other common eye problems and vice versa, and highlights the need to manage all eye problems appropriately alongside cysteamine treatment, recommending when treatments can be used concomitantly and when cysteamine therapy should be deferred.Understands the importance of compliance in delivering positive outcomes and provides advice on how to deliver this at different stages of a person’s life and at different stages of the disease. A focus is to educate patients and encourage and enable topical cysteamine adherence.Utilises a classification system that helps to tailor care and management appropriately.

The goal of the management approach is to reduce the complications of the disease and to keep patients in their current 3C classification category for as long as possible. This ultimately helps to maintain the quality of life of patients by delaying or preventing irreversible ocular complications. In publishing the details of our approach, we aim not only to inform others, but also to encourage the medical community to develop a consensus on evidence-based long-term guidelines for the management of ocular cystinosis.

## Figures and Tables

**Figure 1 ijms-26-08237-f001:**
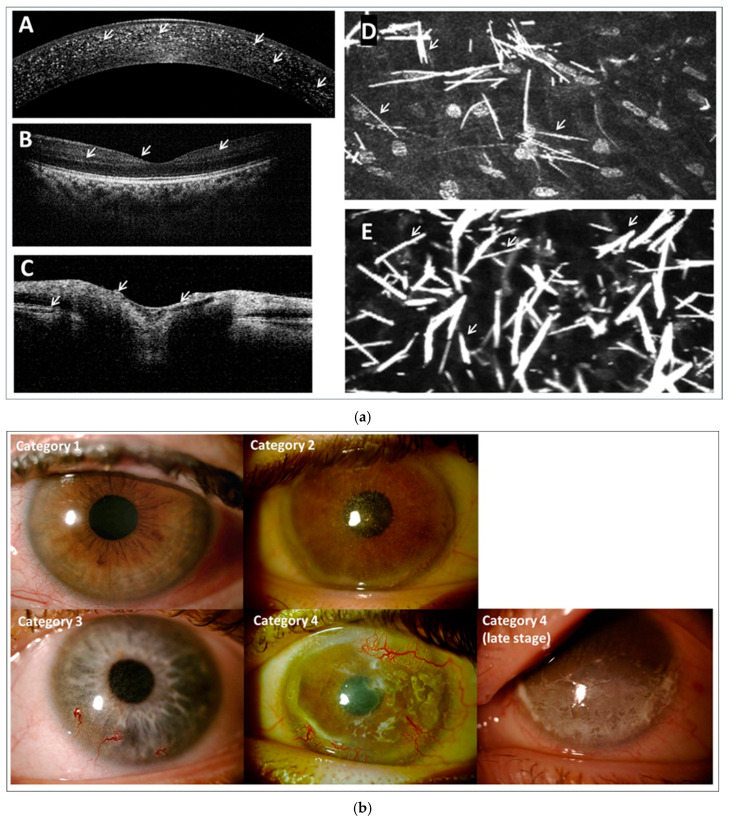
(**a**) Routine imaging of the eye for cystinosis patients. Anterior segment OCT of cystinotic cornea (**A**), posterior segment OCT of cystinotic macula, in a patient well managed and compliant to systemic therapy (**B**), and posterior segment OCT of cystinotic optic nerve (**C**). In Vivo Confocal Microscopy imagery from cystinosis patients in 3C classification Category 1 (**D**) and Category 4 (**E**). White arrows highlight examples of crystal deposits. (**b**) Typical examples of patients in each defined 3C classification Category. Category 1: Patient in middle childhood 6–11 years old, diagnosed before 1 year of age, very few cystine crystals observed. Category 2: Young adult patient 19–21 years old, diagnosed at approximately 1 year of age, cystine crystals abundant in the cornea even at anterior surface of the eyelid. Category 3: Adult patient < 40 years old, diagnosed between 2 and 3 years of age, beginning of corneal neovascularisation observed, cystine crystals widespread across the cornea. Category 4: Adult patient < 40 years old, diagnosed between 2 and 3 years of age, severe complications including the following: keratitis, band keratopathy, neovascularisation, and repeated ulceration with opacification. Category 4 (late stage): Adult patient > 40 years old, very severe cystinosis complications, total opacification with severe crisis of corneal ulcers.

**Figure 2 ijms-26-08237-f002:**
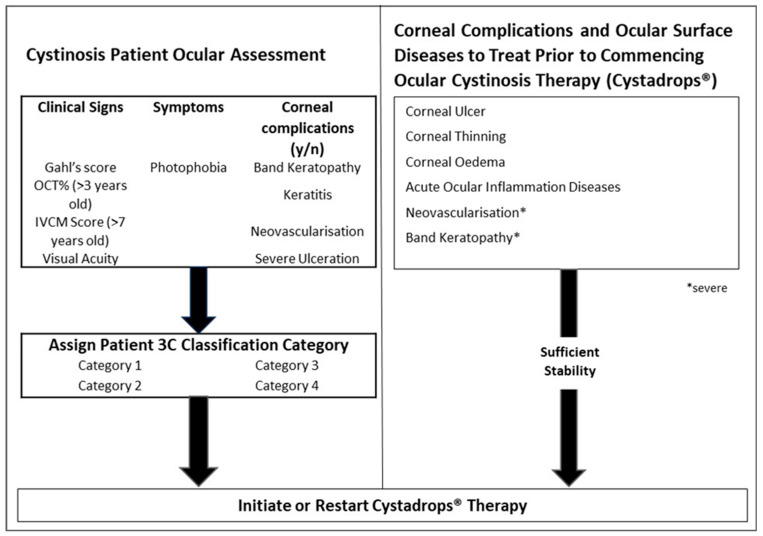
Initiating Cystadrops^®^ therapy.

**Figure 3 ijms-26-08237-f003:**
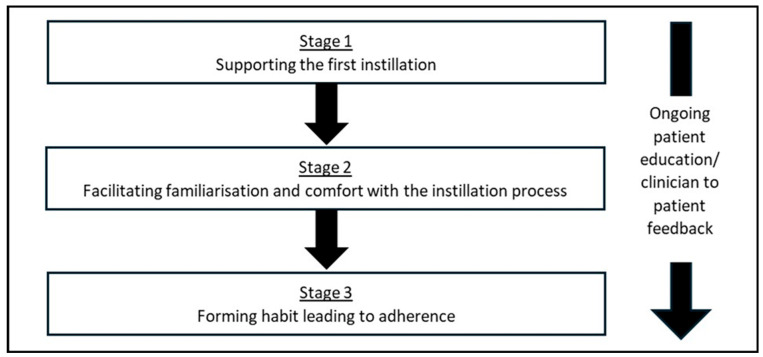
Management approach to encourage topical cysteamine adherence.

**Figure 4 ijms-26-08237-f004:**
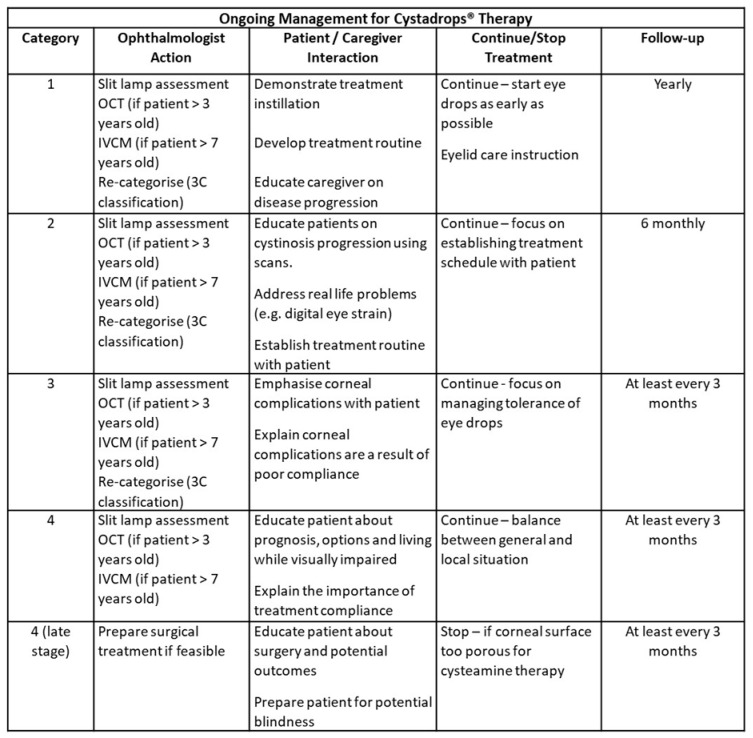
Simplified management approach—ongoing management of ocular cystinosis with Cystadrops^®^ therapy.

**Figure 5 ijms-26-08237-f005:**
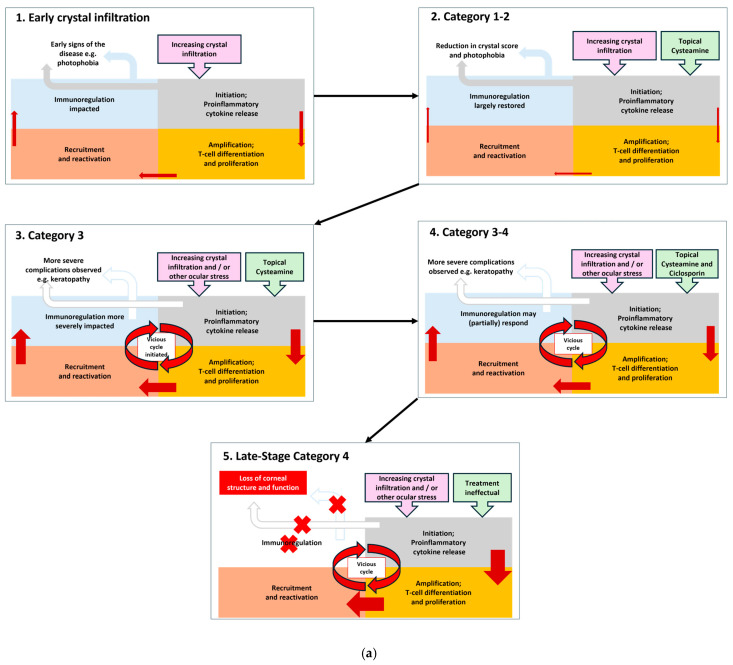
(**a**). Hypothetical schema of the proposed vicious cycle of corneal cystinosis; incorporating crystal formation, inflammation and treatment. Cystinosis in the cornea is initially relatively controlled by innate immunoregulation, however over time this becomes impacted. Crystal load is observed alongside early signs of the disease (1). Initiation of topical cysteamine reduces crystal load and inflammation restoring/partially restoring innate immunoregulation (2). Patients with good compliance to cysteamine can remain in Category 1 or 2 for several decades. Other ocular stresses, inflammatory diseases and/or lack of adherence to cysteamine lead to increasing crystal deposition and increasing inflammation and innate immunoregulation becomes severely impacted and the vicious cycle of cystinosis if left unchecked commences (3). Innate immunoregulation starts to become dysfunctional as indicated by the white arrows. Addition of cyclosporin may allow some response in native immunoregulation (4) keeping patients in Category 3/early-stage Category 4. By late-stage Category 4 the vicious cycle of cystinosis has taken over and inflammation is driving the clinical manifestations of the disease. Innate immunoregulation is dysfunctional as indicated by the red crosses. The corneal structure becomes compromised which paradoxically leads to corneal crystal decline (5). Width of arrows indicates degree of inflammatory disruption. Adapted from Periman et al., 2020 [61] and Sheppard et al., 2023 [62]. (**b**). Clinical aspects of the vicious cycle of ocular cystinosis.

**Figure 6 ijms-26-08237-f006:**
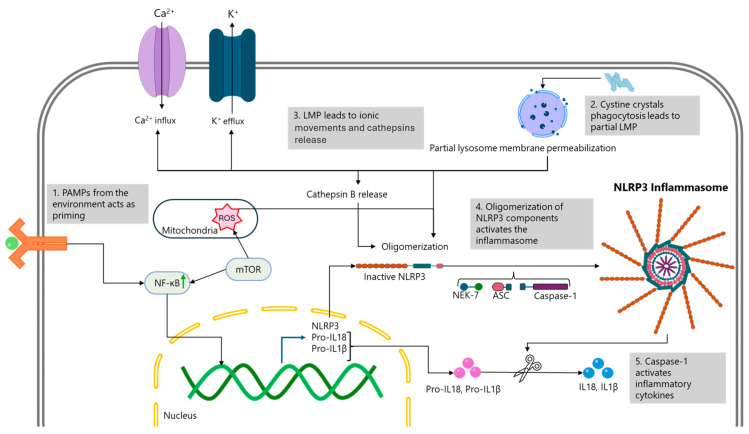
Proposed mechanism of NLRP3 Inflammasome activation in ocular cystinosis [26,43,44,63].

**Table 1 ijms-26-08237-t001:** Routine cystinosis assessment chart as used by the Quinze-Vingts ophthalmology unit.

Frequency of instillation of cysteamine eyedrops *: Tolerance of cysteamine *: Other associated eyedrops *:		Right Eye	Left Eye	
Photophobia (0–5)				
BCVA (0–1)				
Anterior segment—Slit lamp exam	Corneal Cystine Crystal Score (0–3)			
	Keratitis (0–5)			
	Neovascularisation (0–3)			
	Band Keratopathy * (Y/N)			
	Ulceration * (Y/N)			
Anterior segment OCT (0–100%)				
IVCM * (0–28)				
Intraocular pressure				
Posterior segment OCT	Macula			
	Optic nerve			
Visual Field				
Retinography (if necessary)				
Retinal angiography (if necessary)				

* Additional to French recommendations, routine in Quinze-Vingts Hôpital practice. Photophobia is assessed according to Liang’s 5-point cystinosis photophobia scale, where Grade 0 = No photophobia under the slit-lamp beam even with the largest slit; Grade 1 = Photophobia to moderate slit-lamp beam light; Grade 2 = Photophobia to the lightest slit-lamp beam; Grade 3 = Photophobia with inability to tolerate the blue slit-lamp beam; Grade 4 = Photophobia requiring dark glasses. The patients is unable to open the eyes inside the illuminated consultation room. Grade 5 = The patient is unable to open the eyes even inside the dark room. Corneal cystine crystal score is assessed according to Gahl’s score, this score is determined by comparing slit-lamp images of the cornea to a library of images with varying crystal densities with 0 representing the absence of crystals and 3 representing a cornea packed with crystals. Keratitis is graded according to the Oxford staining scale. Neovascularisation grading—Grade 0: No neovascularization is present. Grade 1: There is peripheral invasion of the cornea, typically 1 to 2 mm inside the limbus. Grade 2: The neovascularization is mid-peripheral. Grade 3: The neovascularization involves the entire cornea. Anterior segment OCT%, is recorded according to crystal deposition depth as a percentage of corneal thickness. The IVCM score is determined by calculating the corneal cystine crystal density at the cellular level across 7 layers of the cornea. Images of the superficial epithelium, the basal epithelium, Bowman’s layer, the anterior stroma, middle stroma, posterior stroma, and endothelium are all assessed and scored where 0 = no crystals, 1 = <25% deposits in images, 2 = 25% to 50% deposits in images, 3 = 50% to 75% deposits in images and 4 = 75% to 100% deposits in images. The total out of 28 indicates the severity of crystal deposition.

## Data Availability

Data sharing is not applicable to this article.

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
