# Peer review of "From Molecular Understanding and Pathophysiology to Disease Management; A Practical Approach and Guidance to the Management of the Ocular Manifestations of Cystinosis"

_ijms, 2025, doi:10.3390/ijms26178237_

Round 1
Reviewer 1 Report
Comments and Suggestions for Authors
Major Revision Suggestions:
- Mechanistic Hypothesis Lacks Experimental Validation: The NLRP3 pathway hypothesis (Figure 4) is solely based on literature deduction and lacks experimental validation (â‘ Corneal tissue/cell-level data, e.g., NLRP3 knockout animal models, immunofluorescence of patient corneal biopsies. â‘¡Direct evidence linking cystine crystals to NLRP3 activation (e.g., in vitro crystal stimulation assays).
- Insufficient Evidence for 3C Classification Criteria:The OCT% thresholds (e.g., >50% for Category 3) lack statistical justification.
- Weak Evidence for Anti-Inflammatory Therapy Efficacy: Only one case of cyclosporine combination therapy is described.
- Missing Key Data Tables: Baseline characteristics of all 64 patients (e.g., age distribution, genotype, medication adherence rates).
- Ambiguous Conflict of Interest (COI) Statement:Recordati-funded APC, but authors TD/VG are company employees—potential bias risk. A detailed clarificationis required to state whether corporate sponsorship influenced treatment recommendations.
- Redundant Content in Discussion Section: Repetition of introductory material (e.g., cystinosis definition in §1.2 vs. §3).
- Inconsistent Terminology:Mixed use of "ciclosporin" (INN standard) and "cyclosporine" (US spelling), which should be standardize throughout the manuscript.
- A Treatment Algorithmis Suggested to demonstratewhen to initiate/discontinue ciclosporin.
Author Response
Dear Reviewer - please see the attachment for our response

Reviewer 2 Report
Comments and Suggestions for Authors
(A) Provide an overview/summary of the manuscript
This review delves into cystinosis, a rare lysosomal storage disorder marked by the buildup of cystine crystals within cells, particularly in the cornea, resulting in photophobia and visual impairment. Topical cysteamine remains vital in curbing crystal accumulation. As the disease progresses, inflammation becomes more prominent, especially when management is suboptimal. With improved survival rates, optimizing ocular care is crucial; however, long-term management guidelines are currently lacking. Using the 3C severity assessment system, this paper proposes strategic management strategies aimed at improving patient outcomes and investigates potential molecular pathways involved in corneal inflammation.
(B) Introduction and discussion
The authors emphasized the goals and significance of their research, with conclusions firmly grounded in the data provided.
(C) Review
The review is expertly written, showcasing its reliability and validity.
(D) Reviewer's comment
The authors have conducted a comprehensive review, supported by appropriate references, and the manuscript is written in a scholarly manner.
#1. The French term "mouvement" in Figure 4 should be replaced with the English word "movement" to ensure consistency.
Author Response
Dear Reviewer - thank you for your review, please see the attached file for the full response.

Reviewer 3 Report
Comments and Suggestions for Authors
This is a well-written, but incomplete review of the management of ocular manifestations of cystinosis. The authors understandably focus on the anterior segment of the eye. There is little mention of the posterior eye manifestations of cystinosis, and no mention of systemic cysteamine therapy until the first line of the discussion. This reviewer recommends adding a short section on a description of the posterior eye manifestations of cystinosis along with an explanation for why patients must take two formulations of cysteamine for optimal management of eye disease. Ocular cysteamine drops are required to treat the corneal manifestations of cystinosis because the cornea is avascular and does not provide exposure to systemic cysteamine. Also, the management is strictly about Cystadrops with no mention of oral cysteamine and whether compliance is an issue. If the authors can round out their review by including information on posterior eye manifestations and systemic cysteamine, this will be a thorough review. The following points are recommended edits and a few clarifying questions:
- Line 100. Change inhibited to deficient.
- Line 101. Change mutation to deficiency.
- Line 116. Change distressed to stressed and excrete to secrete.
- Line 136. Change Ctns to Ctns
- Line 145. Change Ctns to Ctns
- Line 150. Change protein Galectin-3 to protein, galectin-3
- Line 154. Change is to was
- Line 184. Change is to was
- Line 191. Change on to of
- Line 192. Change Pathway to pathway
- Line 217. Please clarify what >30% refers to, e.g., crystal deposition depth as a percentage of corneal thickness
- Line 224. Please define the extremes of the scale and provide a reference
- Line 239. Change target to goal, and change key to 3C
- Line 240. Change “. An objective of management is to try “and”
- Line 241. Change “particularly from Category 3 to Category 4” to “, particularly from Category 3 to Category 4,”
- Line 265. Change “and here” to “, where”
- Line 267. Change increased to increases, and change category to Categories
- Table 1. Add a caption explaining the scores on the four scales.
- Line 335. Change category to Category, and change IVCM to In vivo confocal microscopy
- Line 339. Change Late adolescent to Young adult
- Line 343. Describe the Gahl’s score.
- Line 353. Change micros-copy to microscopy
- Line 354. Change rec-ognised to recognised
- Line 372. Adherence and compliance are different treatment concepts that are worth educating the reader because medication use is such an important variable affecting disease progression in cystinosis. The authors could start this paragraph by defining adherence and compliance, e.g., “Adherence refers to a patient actively choosing to follow a mutually agreed-upon treatment plan, reflecting shared decision-making and patient autonomy. Compliance refers to a patient passively taking a medication exactly as the provider prescribes, reflecting a more paternalistic model with the patient following orders. Compliance can suggest blame if the patient doesn’t follow instructions, whereas non-adherence considers barriers to treatment, e.g., cost, side effects, or understanding.” The authors could then continue with their text as written.
- Line 414. Change category to Category
- Line 423. Add the pathophysiology of before cystinosis.
- Line 448. Change objective to purpose. Objective answers how something is done, whereas purpose answers why something is done.
- Line 457. Change follow-ups to follow-up visits
- Line 469. Change three to 3
- Libe 475. “3-monthly follow-up” is unclear to this reviewer. Do the authors mean monthly follow-up for 3 months, or follow-up every 3 months?
- Line 488. Change category to Category
- Line 494. Change categories to Categories
- Line 495. Change category to Category
- Line 509. Remove second present
- Line 542. A word is missing after relatively. controlled?
- Line 624. Change oc-ular to ocular
This is a well-written, but incomplete review of the management of ocular manifestations of cystinosis. The authors understandably focus on the anterior segment of the eye. There is little mention of the posterior eye manifestations of cystinosis, and no mention of systemic cysteamine therapy until the first line of the discussion. This reviewer recommends adding a short section on a description of the posterior eye manifestations of cystinosis along with an explanation for why patients must take two formulations of cysteamine for optimal management of eye disease. Ocular cysteamine drops are required to treat the corneal manifestations of cystinosis because the cornea is avascular and does not provide exposure to systemic cysteamine. Also, the management is strictly about Cystadrops with no mention of oral cysteamine and whether compliance is an issue. If the authors can round out their review by including information on posterior eye manifestations and systemic cysteamine, this will be a thorough review. The following points are recommended edits and a few clarifying questions:
- Line 100. Change inhibited to deficient.
- Line 101. Change mutation to deficiency.
- Line 116. Change distressed to stressed and excrete to secrete.
- Line 136. Change Ctns to Ctns
- Line 145. Change Ctns to Ctns
- Line 150. Change protein Galectin-3 to protein, galectin-3
- Line 154. Change is to was
- Line 184. Change is to was
- Line 191. Change on to of
- Line 192. Change Pathway to pathway
- Line 217. Please clarify what >30% refers to, e.g., crystal deposition depth as a percentage of corneal thickness
- Line 224. Please define the extremes of the scale and provide a reference
- Line 239. Change target to goal, and change key to 3C
- Line 240. Change “. An objective of management is to try “and”
- Line 241. Change “particularly from Category 3 to Category 4” to “, particularly from Category 3 to Category 4,”
- Line 265. Change “and here” to “, where”
- Line 267. Change increased to increases, and change category to Categories
- Table 1. Add a caption explaining the scores on the four scales.
- Line 335. Change category to Category, and change IVCM to In vivo confocal microscopy
- Line 339. Change Late adolescent to Young adult
- Line 343. Describe the Gahl’s score.
- Line 353. Change micros-copy to microscopy
- Line 354. Change rec-ognised to recognised
- Line 372. Adherence and compliance are different treatment concepts that are worth educating the reader because medication use is such an important variable affecting disease progression in cystinosis. The authors could start this paragraph by defining adherence and compliance, e.g., “Adherence refers to a patient actively choosing to follow a mutually agreed-upon treatment plan, reflecting shared decision-making and patient autonomy. Compliance refers to a patient passively taking a medication exactly as the provider prescribes, reflecting a more paternalistic model with the patient following orders. Compliance can suggest blame if the patient doesn’t follow instructions, whereas non-adherence considers barriers to treatment, e.g., cost, side effects, or understanding.” The authors could then continue with their text as written.
- Line 414. Change category to Category
- Line 423. Add the pathophysiology of before cystinosis.
- Line 448. Change objective to purpose. Objective answers how something is done, whereas purpose answers why something is done.
- Line 457. Change follow-ups to follow-up visits
- Line 469. Change three to 3
- Libe 475. “3-monthly follow-up” is unclear to this reviewer. Do the authors mean monthly follow-up for 3 months, or follow-up every 3 months?
- Line 488. Change category to Category
- Line 494. Change categories to Categories
- Line 495. Change category to Category
- Line 509. Remove second present
- Line 542. A word is missing after relatively. controlled?
- Line 624. Change oc-ular to ocular
Author Response

(The authors gave the same response as above.)

Round 2
Reviewer 1 Report
Comments and Suggestions for Authors
A section on limitations and long-term safety should be added at the end of the discussion.
